# Proteome Alterations in Equine Osteochondrotic Chondrocytes

**DOI:** 10.3390/ijms20246179

**Published:** 2019-12-07

**Authors:** Elisabetta Chiaradia, Marco Pepe, Pier Luigi Orvietani, Giovanni Renzone, Alessandro Magini, Monica Sforna, Carla Emiliani, Antonio Di Meo, Andrea Scaloni

**Affiliations:** 1Department of Veterinary Medicine, University of Perugia, 06126 Perugia, Italy; marco.pepe@unipg.it (M.P.); monica.sforna@unipg.it (M.S.); antonio.dimeo@unipg.it (A.D.M.); 2Department of Experimental Medicine, University of Perugia, 06132 Perugia, Italy; pier.orvietani@unipg.it; 3ISPAAM, National Research Council, 80147 Naples, Italy; giovanni.renzone@ispaam.cnr.it (G.R.); andrea.scaloni@ispaam.cnr.it (A.S.); 4Department of Chemistry, Biology and Biotechnology, University of Perugia, 06123 Perugia, Italy; alessandro.magini@unipg.it (A.M.); carla.emiliani@unipg.it (C.E.); 5CEMIN-Center of Excellence for Innovative Nanostructured Material, 06123 Perugia, Italy

**Keywords:** chondrocytes, osteochondrosis, proteomics, joint diseases, human juvenile osteochondrosis, endochondral ossification, growth plate and cartilage development, horse

## Abstract

Osteochondrosis is a failure of the endochondral ossification that affects developing joints in humans and several animal species. It is a localized idiopathic joint disorder characterized by focal chondronecrosis and growing cartilage retention, which can lead to the formation of fissures, subchondral bone cysts, or intra-articular fragments. Osteochondrosis is a complex multifactorial disease associated with extracellular matrix alterations and failure in chondrocyte differentiation, mainly due to genetic, biochemical, and nutritional factors, as well as traumas. This study describes the main proteomic alterations occurring in chondrocytes isolated from osteochondrotic cartilage fragments. A comparative analysis performed on equine osteochondrotic and healthy chondrocytes showed 26 protein species as differentially represented. In particular, quantitative changes in the extracellular matrix, cytoskeletal and chaperone proteins, and in cell adhesion and signaling molecules were observed in osteochondrotic cells, compared to healthy controls. Functional group analysis annotated most of these proteins in “growth plate and cartilage development”, while others were included in “glycolysis and gluconeogenesis”, “positive regulation of protein import”, “cell–cell adhesion mediator activity”, and “mitochondrion nucleoid”. These results may help to clarify some chondrocyte functional alterations that may play a significant role in determining the onset and progression of equine osteochondrosis and, being related, of human juvenile osteochondrosis.

## 1. Introduction

Osteochondrosis (OC) is a multifocal disease, which affects articular-epiphyseal cartilage complexes and growth plates in various mammalian species such as horses and humans. The cause of OC is not fully understood, even if different factors, including skeletal growth rates, nutrition, genetics, or physical activity, have been implicated [1,2,3,4,5,6]. The pathogenesis of OC is similar across the species [7]. In humans, osteochondroses are a group of disorders with a different name according to the affected anatomical sites. In particular, equine OC has similar symptoms, predilection sites, and clinical presentation as human juvenile OC [7]. In both humans and horses, OC is characterized by focal chondronecrosis and retention of growth cartilage, which can lead to the formation of fissures, subchondral bone cysts, and intra-articular fragments (osteochondrosis dissecans or OCD) [2,3]. This disorder is one of the most common causes of degenerative joint diseases. Indeed, the progression of cartilage lesions and the presence of intra-articular fragments can also develop in early onset osteoarthritis (OA) [4,5]. The molecular events that can drive the chondrocyte biogenesis impairment and extracellular matrix (ECM) alteration, common features in OC, seem to be related to mitochondrial and endoplasmic reticulum alterations, oxidative stress, and endocrinological dysfunction [8].

As for other joint diseases, interactions between chondrocytes and ECM may play key roles in OC. Indeed, chondrocytes, the only cell type present in articular cartilage, represent the main coordinators of endochondral ossification. They synthetize molecular components of ECM and the enzymes involved in the normal cartilage turnover. However, alterations in molecular ECM composition represent further signals for modulating chondrocyte metabolism [9].

In this study, differential gel electrophoresis analysis followed by mass spectrometry-based protein identification were performed in order to determine the protein alterations that occur in equine OC chondrocytes, and to clarify molecular processes and metabolic aspects of damaged tissues. In recent years, proteomics has been used to investigate the pathogenesis of joint diseases such as OA and rheumatoid arthritis [10,11,12,13]. However, there are few proteomic studies on osteochondrosis, and most of them are focused on tissue (cartilage and subchondral bone) [14,15] or synovial fluid [16,17] analysis. To the best of our knowledge, this work describes the first proteomic analysis of osteochondrotic chondrocytes and corresponding molecular signatures, which could be indicative of the alterations of cell functions involved in osteochondrosis. This study also provides suggestions for new research topics and for clarifying the molecular aspects of OC across different animal species.

## 2. Results

### 2.1. 2DE Analysis of Proteins Extracted from OC and Normal Chondrocytes

Proteomic profiles of chondrocytes isolated from normal (Controls) and osteochondrotic (OC) cartilage were analyzed using 2DE combined with mass spectrometry (MS) procedures in order to identify protein signatures, which may be indicative of the molecular alterations occurring in the disease. The analyses were performed by comparing protein spots resolved in 2DE gel within p*I* 3–10 and M*r* 10–150 kDa ranges (Figure 1). Coomassie staining evidenced an average of 188 ± 20 and 177 ± 12 spots in CTR-gels and OC-gels, respectively, with a degree of similarity 88.91%.

Quantitative comparative analysis evidenced 64 peptide spots (marked in Figure 1) having a representation variation greater than twofold in OC gels, with respect to controls (*p* < 0.05). They corresponded to 26 different gene products. These protein spots and corresponding fold changes are reported in Table 1. The protein identification and MS analysis data are described in Appendix A.

Thirty-one spots corresponded to three alpha chains of collagen VI, namely collagen alpha-1 (VI) chain (COL6A1), collagen alpha-2 (VI) chain (COL6A2), and collagen alpha 3 (VI) chain (COL6A3). More specifically, we observed a decrease in spots with similar experimental and theoretical molecular mass values, while spots identified as probable fragments (experimental molecular mass < theoretical molecular mass) were highly abundant. Some of these fragments were only detected in the OC gels. Other deregulated ECM and perichondrium proteins were hyaluronan and proteoglycan link protein 1 (HAPLN1) and cartilage intermediate layer protein 1 (CILP-1). The proteomic analysis also showed alterations in cytoskeletal proteins, such as vimentin (VIM), junction plakoglobin (JUC), and actin cytoplasmic 1 (ACTG1), as well as in chaperone proteins. In particular, increases were observed for endoplasmic reticulum resident protein 44 (ERp44), while decreases were observed for endoplasmin (HSP90B1) and 14-3-3 protein theta (YWHAQ). There was also a down-representation of enzymes involved in amino acid metabolism, such as mitochondrial ornithine aminotransferase (OAT) and adenosylhomocysteinase (AHCY).

The OC cells also showed over-representation of glycolytic enzymes, namely triosephosphate isomerase (TPI1), enolase 1 (ENO1), and L-lactate dehydrogenase (L-LDH), as well as the enzymes involved in the mitochondrial energy production, such as ATP synthase beta subunit (ATP5B), succinyl-CoA ligase (SUCLG1), trifunctional mitochondrial enzyme subunit beta (HADHB), and enoyl-CoA hydratase mitochondrial (ECHS1). Moreover, the osteochondrotic cells showed increases in voltage-dependent anion-selective channel protein 2 (VDAC2), lactadherin (MFGE8), annexin A1 (ANXA1), and annexin A2 (ANXA2). Lastly, three spots with different fold change trends corresponding to superoxide dismutase (SOD) were also identified.

### 2.2. Protein Enrichment Analysis

The protein–protein interaction analysis performed using STRING (Figure 2A) indicated that the proteins differentially represented in OC cells are highly functionally correlated (*p* < 1.0 × 10^−16^), and none of our proteins proved to be individual proteins without connections to others. The extracted interactome obtained was comprised of 26 nodes and 78 edges, while the average node degree and average local clustering coefficient of the network were 6 and 0.723, respectively. The gene ontology (GO) category cellular component associated 12 proteins to extracellular regions (GO:005576), 9 to mitochondrion (GO:005739), and 13 to Cytosol (GO:005829), which are shown in Figure 2A, in green, red, and blue color, respectively. The complete cell component annotation, including the 42 significantly enriched GO terms, proteins, and false discovery rate (FDR), are listed in Appendix A. Panther analysis identified the most represented protein classes (Figure 2B), namely chaperone, transporter, hydrolase, oxidoreductase, cell adhesion molecule, lyase, transferase, nucleic acid binding, ligase, receptor, isomerase, cytoskeletal protein, signaling molecules, and extracellular matrix proteins.

The two Cytoscape plugins, ClueGO and the CluePedia, were used to investigate the functionally grouped networks of deregulated proteins in OC in detail. In particular, biological processes, molecular functions, cellular component, Reactome, Kyoto Encyclopedia of Genes annotations and Genomes (KEGG), and WikiPathways were integrated. The κ score level was set at ≥0.4. The statistically significant (*p* < 0.05) enrichment terms are shown in Figure 3A. The most significant functional groups evidenced were glycolysis and gluconeogenesis, growth plate and cartilage development, positive regulation of protein import, cell–cell adhesion mediator activity, and mitochondrion nucleoid. All functional groups/pathway terms and the relative percentage (%) of deregulated genes in OC are shown in Figure 3B.

### 2.3. Validation of Proteomics Results

In order to validate quantitative variations evidenced by the proteomic analysis, various approaches were used. More specifically, immunoblotting was performed for VIM, ANX1 and ANX2, and collagen V alpha 1 chain (Figure 4A), which confirmed the results obtained by the comparative densitometric analysis of 2DE gel spots. Interestingly, no changes were observed when tubulin (TUB) was immunodetected, excluding the involvement of this protein in OC cell cytoskeleton alterations; therefore, the Western Blotting and of TUB was used as a housekeeping gene for the densitometric analysis of other protein bands. VIM, ANX1, and ANX2 were chosen to validate the quantitative changes impacting alterations of cytoskeleton, also assayed by florescence analysis, occurring in OC cells. Definitively, variations in actin cytoskeleton were confirmed by staining the cells with fluorescent-labelled phalloidin (Figure 4B), which showed a significant increase in OC cell signals. The decrease in COL6A1, indicating alterations in the perichondrium, was also confirmed by the immunohistochemical evaluation of OC and the control cartilage through the use of specific antibodies (Figure 4C). Indeed, the control cartilage showed an extensive and high pericellular expression of collagen VI, while a significant decrease in pericellular immunoreaction was observed in the OC samples.

## 3. Discussion

In this study, a proteomic approach was used to clarify the functional alterations that occur in osteochondrotic chondrocytes. The identification of deregulated proteins, their location, and the pathways in which they are involved could enhance our knowledge about the pathogenic processes related to OC. More specifically, a gel-based proteomic analysis was carried out, followed by spot identification with mass spectrometry procedures in order to detect changes in protein abundance in OC cells. To gain a better understanding of the biological relevance of deregulated proteins, bioinformatics tools were also used and validation tests were performed to confirm the results.

Some proteins found as differentially represented in OC cells were annotated to the functional term “growth plate and cartilage development”. It is worth noting that OC is a multifactorial disorder that affects the growing skeleton and it is characterized by abnormal cartilage and subchondral bone formation, for which our results may prove useful for identifying key protein players of OC pathogenesis.

Proteomic analysis of OC cells identified proteins that may be responsible for the ECM alterations characterizing OC cartilage, such as the HAPLN1 and CILP-1. HAPLN1 plays a key role in stabilizing the ECM as it links the aggregates of aggrecan and hyaluronic acid, and acts as a growth factor by stimulating the synthesis of aggrecan and type II collagen [18,19]; CILP-1 proved to be essential for cartilage structure and scaffolding [20]. Synthesis of both these proteins is modulated by SOX-9, (key modulator of chondrogenesis and synthesis of cartilage proteins) [21], which is altered in OC [22]. In agreement with to a histological study [23], our results also revealed changes in the pericellular matrix that surrounds the chondrocytes, due to a global decrease in COL VI chains, probably induced by high proteolytic activity, which has been well-documented in literature [24] and biosynthesis defects. Protein changes associated with proteolytic degradation should eventually have escaped detection whenever gel-free shotgun proteomic procedures should have been used. Indeed, we also found low levels of OAT, the key enzyme involved in biosynthesis of proline, which is the most abundant amino acid in collagen. A reduction in COLVI can lead to a decrease in cartilage elasticity and flexibility in the adaptation to mechanical stress-induced and load-induced changes [25,26]. In addition, disruption of the pericellular matrix has consequences in important processes involved in development, maintenance, and degeneration of articular cartilage, which modulate the organization of the growth plate [27].

In accordance with Desjardin and coworkers [14], our results revealed cytoskeletal alterations, which play an important role in maintaining the chondrogenic phenotype and structural integrity, and are essential for collagen transport and secretion [28,29]. In particular, we observed changes in ACTG1 and VIM, which were confirmed by fluorescence analysis using labeled phalloidin and Western blotting, respectively. Both proteins are involved in chondrogenesis and chondrocyte differentiation [30,31], in determining mechanical properties, as well as in regulating the response of chondrocytes to compressive loading conditions [32]. Low levels of VIM have been reported for other skeletal system diseases [33,34]. Moreover, it has been demonstrated that cytoskeletal alterations can induce chondrocyte apoptosis, ECM alterations, downregulating aggrecan and collagen II expression, and inducing matrix metallopeptidases (MMP) synthesis [35]. Cytoskeletal alterations may also be due to changes in the levels of JUP (a junctional plaque protein), which plays an important role in intercellular junctions and ECM adhesions. It binds to cell adhesion molecules and transmembrane components of the intermediate filaments [36], modulating the cell–matrix interactions that are essential for chondrogenesis [37]. JUP is directly involved in the arrangement and function of the cytoskeleton and the cells within the tissue. Thus, as indicated by the bioinformatic analysis, cell adhesion seems to be highly compromised in OC. Further modifications of the cytoskeleton, plasma membrane, and extracellular matrix interactions may be due to increased AnxA1 and AnxA2 levels, which are both involved in actin cytoskeletal rearrangements and/or extracellular proteins [38]. Moreover, AnxA1 and AnxA2 are involved in controlling cellular proliferation, differentiation, migration, and apoptosis [39,40,41]. Similar effects on cytoskeletal structure and cell adhesion may be due to changes in endoplasmic reticulum proteins such as YWHAQ, endoplasmin, and ERp44. More specifically, YWHAQ (also named 14-3-3 theta protein) is an adapter protein that binds protein kinases, receptor proteins, cytoskeleton, and scaffolding proteins [42,43]. Endoplasmin, also called HSP90B1, gp96 or grp94, is known to be involved in chondrocyte differentiation and growth plate vascularization [43]. Moreover, it assists in the folding of growth signaling and cell adhesion proteins [44]. In fact, chaperones play a key role in endochondral ossification and cartilage pathology [45]. The alterations in chaperone levels, in addition to the impairment in some amino acid’s metabolism, such as proline, previously mentioned, and sulfur amino acids [46], due to the low levels of AHCY found in OC cells, could be also indicative of alteration of protein quality. Moreover, impairment in transsulfuration pathways may have consequences in epigenetic modulation of gene expression, glutathione, and taurine synthesis, as well as in proteoglycan sulfation, which is crucial for cartilage and bone homeostasis, and remodeling [47,48,49,50,51,52].

These results also showed a general increase in cytosolic (TPI1, GAPDH, ENO1), L-LDH, and mitochondrial (SUCLG1, HADHB and ECHS1, ATP synthase) enzymes, which are involved in energy production. These findings could be interpreted as an adaptative response to the reduction of nutrient supply, which occurs in OC fragments, where chondrocytes are far from the capillaries originated from the epiphyseal artery, the only source of glucose for growth plate cartilage [53].

Finally, some of our results sustain the hypothesis that the increased apoptosis of chondrocytes may play a role in OC [54]. Most of the previously discussed protein changes, including chaperones ANAX1-2, COL VI, and cytoskeleton proteins, suggested the occurrence of proapoptotic conditions in OC cartilage [35,55,56], which could be worsened by the increase of VDAC2, also known as the mitochondrial porin protein, which is a key molecule in mitochondria-mediated apoptosis [57,58].

Taken together, these results reveal the molecular signatures of OC chondrocytes that could be indicative of alterations in cell functions associated with osteochondrosis. In particular, this study provides insights into the role of deregulated proteins and describes the pathways in which they are involved. Most of these pathways are related to chondrocyte and bone development processes, as well as to chondrogenesis, thus confirming that the proteins we found can trigger endochondral ossification that failed in OC. Since horse disease is the best animal model of human juvenile OC, this study also provides suggestions for new research topics and for clarifying the molecular aspects of OC across the species.

## 4. Materials and Methods

### 4.1. Cartilage Sampling and Processing

Equine chondrocytes were isolated from macroscopically normal healthy articular cartilage (Control) and from osteochondritic fragments (OC) of the sagittal ridge of the metacarpus/metatarsus and dorsoproximal aspect of the proximal phalanx for metacarpo/metatarsophalangeal joints and from the lateral trochlear ridge of femur for femoropatellar joints; these samples were obtained from 14 male thoroughbred horses aged 1–4 years old, 7 with OC and 7 healthy (Control), respectively. Pathological samples were obtained during diagnostic arthroscopy from clinical cases referred to the University of Perugia Veterinary Teaching Hospital, following guidelines of the Animal Care and Use Committee of the University of Perugia. Owners of clinical patients signed waivers for their samples to be used in this study. OC diagnosis was obtained from radiographic and arthroscopic findings by revealing the presence of osteochondral fragments of metacarpo/metatarsophalangeal (sagittal ridge of the third metacarpus/metatarsus) and femoropatellar (lateral trochlear ridge of femur) joints. Control samples were obtained from animals euthanized for reasons other than orthopedic diseases. The joints were selected based on the absence of macroscopic abnormalities/changes consisting of OC or osteoarthritic (OA) lesions. Representative examples of OC fragment and normal cartilage are reported in Figure 5.

Chondrocytes were obtained from cartilage slices harvested using sterile scalpels, according to Mancini and coworkers [59]. Cell pellets were solubilized by vortexing for 3 h in a buffer (2DE lysis buffer) containing 8 M urea, 2 M thiourea, 4% CHAPS, 40 mM Tris, and 100 mM DTT. Protein concentration was estimated by using the Bradford assay.

### 4.2. 2D-Gel Electrophoresis

Six hundred µg of proteins in 2DE lysis buffer (added with 0.8% ampholytes and a trace of bromophenol blue) were used to dehydrate Immobiline Dry Strips (IPG 18 cm, linear 3–10 pH range, Bio-Rad, Hercules, CA, USA). Isoelectrofocusing was performed using a Protean IEF Cell (Bio-Rad, Hercules, CA, USA) at 20 °C; low initial voltage was followed by a voltage gradient from 10,000 to 95,000 Vh, with a limiting current of 50 mA/strip. After focusing, proteins were reduced with 2% *w*/*v* DTT, and then alkylated with 2.5% *w*/*v* iodoacetamide in equilibration buffer (375 mM Tris–HCl pH 8.8, 6 M urea, 20% *w*/*v* glycerol, 2% *w*/*v* SDS). The second dimension was carried out in 12% T gradient slab polyacrylamide gels. 2DE gels were stained with colloidal Coomassie G250 and the resulting images were acquired by using a GS-800 imaging system (Bio-Rad, Hercules, CA, USA). Digitized images of the stained gels were analyzed by using the PDQuest (ver 7.4) 2D analysis software (Bio-Rad, Hercules, CA, USA), which allowed spot detection, landmark identification, aligning/matching of spots within gels, quantification of matched spots, and their analysis, according to manufacturer′s instructions. Manual inspection of the spots was performed to verify the accuracy of automatic gel matching; any errors in the automatic procedure were manually corrected prior to the final data analysis. The spot volume was used as the analysis parameter for quantifying protein expression. The “total quantity in valid spots” was used as normalization mode. 2DE maps were grouped together by using the “replicate group” function. Fold changes in protein spot levels were calculated as ratio of average of spot value in OC vs. those of Control. Spots exhibiting a fold change ≥2 or ≤0.5 were further considered for statistical analysis (*t*-test analysis) performed by R program, version 2.13.2. (R Development Core Team, Vienna, Austria). Spots showing a *p* value <0.05 were considered as significantly different in OC vs. Control groups, and were further analyzed by mass spectrometry for protein identification.

### 4.3. Protein Digestion and MS Analysis

Spots from 2DE were manually excised from gels, triturated, and washed with water. Proteins were in-gel reduced, *S*-alkylated, and digested with trypsin, as previously reported [60]. Protein digests were subjected to a desalting/concentration step on μZipTipC18 pipette tips (Millipore, Bedford, MA, USA). The obtained peptide mixtures were then analyzed by nanoLC-ESI-LIT-MS/MS using an LTQ XL mass spectrometer (Thermo Fisher, Foster City, CA, USA) equipped with a Proxeon nanospray source connected to an Easy-nanoLC (Proxeon, Odense, Denmark) [61]. Peptide samples were separated on an Easy C18 column (100 × 0.075 mm, 3 μm) (Proxeon, Odense, Denmark) using a gradient of acetonitrile containing 0.1% *v*/*v* formic acid in aqueous 0.1% *v*/*v* formic acid. Briefly, acetonitrile ramped from 5% to 35% over 15 min and from 35% to 95% over 2 min, at a flow rate of 300 nL/min. Mass spectra were acquired in the range *m*/*z* 400–2000. Acquisition was performed by a data-dependent product ion scanning procedure over the three most abundant ions, enabling dynamic exclusion (repeat count 2 and exclusion duration 1 min). Mass isolation window and collision energy were set to *m*/*z* 3 and 35%, respectively. MASCOT software package version 2.3.02 (Matrix Science, London, UK) [62] was used for unambiguous protein identification by searching MS data against a nonredundant database of *Equus caballus* protein sequences downloaded from UniProtKB (09/2019). A mass tolerance value of 2 Da for precursor ion and 0.8 Da for MS/MS fragments, trypsin as proteolytic enzyme, a missed cleavage maximum value of 2, and Cys carbamidomethylation and Met oxidation as fixed and variable modification, respectively, were set as MS data search parameters. Protein candidates with at least 2 assigned peptides with an individual MASCOT score >25, corresponding to *p* < 0.05 for a significant identification, were further evaluated by the comparison of their calculated Mr and pI values with the experimental ones from 2DE.

### 4.4. Protein Enrichment Analysis

STRING (http://string-db.org/), Panther (Protein ANalysis THrough Evolutionary Relationships) (http://www.pantherdb.org/), and the Cluego + Clupedia plugin [63,64] of Cytoscape platform version 3.7.2 (https://cytoscape.org) were used to perform GO annotation, PPI enrichment, and integrative analysis.

### 4.5. Validation of Proteomics Results

#### 4.5.1. Western Blotting

Different OC (6) and healthy cartilage (6) samples, obtained with the same conditions and from the same kind of joints used for proteomics, were used to extract cells to validate proteomic results. To this aim, cells were lysated using Ripa Buffer (50 mM Tris-HCl pH 8.0, 250 mM NaCl, 1% NP-40, 0.1% SDS, 1% Na-deoxycholate). The extracted proteins were mixed in three pools and used to perform immunoblotting. Briefly, proteins were separated in SDS-PAGE (10% or 12% T, according to the molecular mass of protein under evaluation) and blotted on Polyvinylidene Difluoride (PVDF) membranes. The immunodetection of proteins was performed incubating blotted membrane with Ab-COL6A1 (Santa Cruz, Dallas, Texas, USA, 1:500), Ab-Annexin II (Santa Cruz, Dallas, Texas, USA, 1:1000), Ab-Annexin I (Santa Cruz, Dallas, Texas, USA, 1:1000), and Ab-Vimentin (Cell Signaling Technology, Leiden, The Netherlands, 1:1000). Detection was carried out using the appropriate secondary antibody, i.e., horseradish peroxidase-conjugated secondary antibody anti-mouse (Sigma-Aldrich Corp., MO, USA, 1:5000) or anti-rabbit (Sigma-Aldrich Corp., MO, USA, 1:5000), based on the first antibody. Immunoreactive proteins were finally evidenced by chemiluminescence using the ECL system and film images were then acquired using a GS-800 imaging systems scanner. Densitometric analysis was performed by QuantityOne 4.5.0 (BioRad, Hercules, CA, USA) using tubulin as normalization factor for band optical density. *t*-tests were used for statistical analysis; *p* < 0.05 was deemed as significant.

#### 4.5.2. Actin Labeling by Fluorescently Labeled Phalloidin

Control and OC cells, isolated as before referred [59], were seeded on glass slips at a density of 4 × 10^4^ cells cm^2^ in six-well plates and incubated at 37 °C in a humidified atmosphere of 5% CO_2_ for 24 h. Cells were then washed twice with PBS and fixed with 4% (*v*/*v*) paraformaldehyde for 20 min at room temperature, washed three times with D-PBS, and blocked in D-PBS containing 5% (*v*/*v*) FBS and 0.3% (*v*/*v*) Triton X-100 for 1 h at RT. The cells were washed three times with D-PBS and incubated for 1 h in Alexa FluorVR 488 Phalloidin (Thermo Fisher Scientific, Waltham, MA, USA, 1:1000). After washing, the cells were counterstained with DAPI. Fluorescence microscopy analysis was performed using a Nikon TE2000 microscope (Nikon Instruments Spa, Florence, Italy) through a 60× oil immersion objective. Fluorescence intensity analysis was achieved by using ImageJ software (NIH). *T*-test was used for statistical analysis, *p* < 0.05 was deemed as significant.

#### 4.5.3. Immunohistochemical Evaluation of COL6A1 in Cartilage

Cartilage obtained from healthy joints (Control) and OC fragments (OC) were embedded in Cryomatrix embedding resin (Shandon) and frozen in isopentane cooled in liquid nitrogen. Tissue samples were stored at −80 °C before sectioning on a Leica CM1900 Cryostat into slices of 5 µm, subsequently mounted on poly-L-lysine–coated slides, and stored at −80 °C. Serial cryosections were warmed to room temperature and incubated for 2 h with antibody against COL6A1 (Santa Cruz Biotechnology, Dallas, Texas, US, 1:100). Slides were then treated with an ABC ready-to-use kit (Abcam, Cambridge, UK) following the manufacturer’s instructions. Positive reaction was revealed with 3-amino-9-ethilcarbazole (Abcam, Cambridge, UK); Mayer’s hematoxylin was applied as a counterstain.

## Figures and Tables

**Figure 1 ijms-20-06179-f001:**
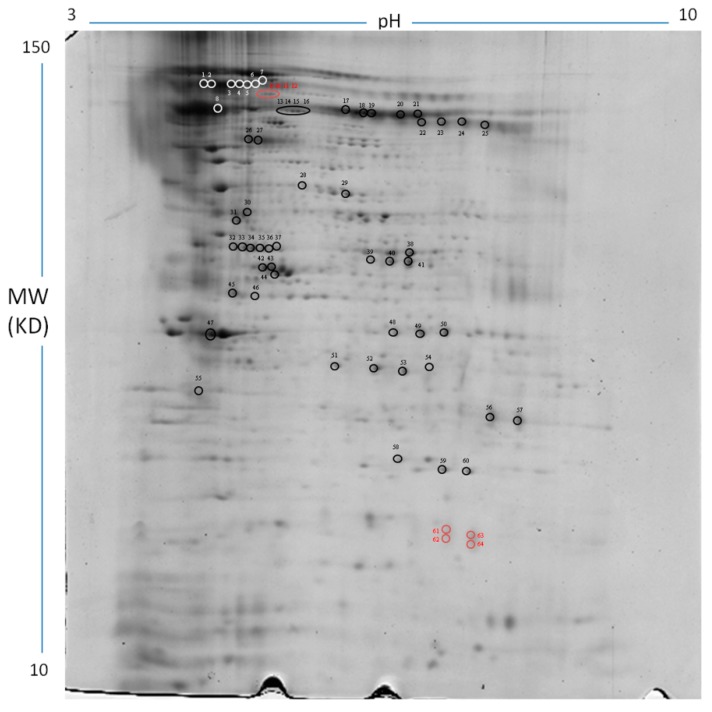
Representative image of a 2D gel of equine chondrocytes obtained using linear IPG (immobilized pH gradient) strip 3–10 and 12% T SDS-PAGE. The differentially represented protein spots (fold change > 2 and *p* < 0.05) were marked and numbered. Corresponding identified proteins are listed in Table 1. Spots found in OC and Control gels are marked in black and white circles, while spots found in OC gels only are marked in red circles.

**Figure 2 ijms-20-06179-f002:**
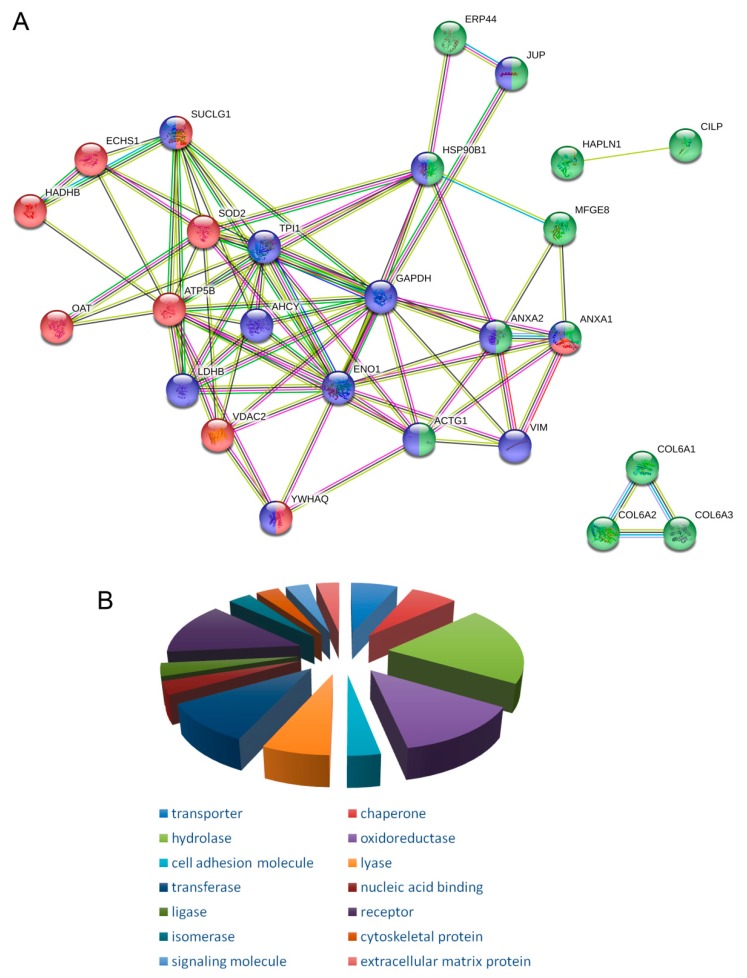
(**A**) STRING Protein Protein Interaction (PPI) analysis of differentially expressed proteins in OC, nodes colors are in accordance with cell component categories: green = extracellular regions (GO:005576); red = mitochondrion (GO:005739); blue = cytosol (GO:005829); (**B**) PANTHER analysis classified the proteins identified according to the corresponding “Protein Class”.

**Figure 3 ijms-20-06179-f003:**
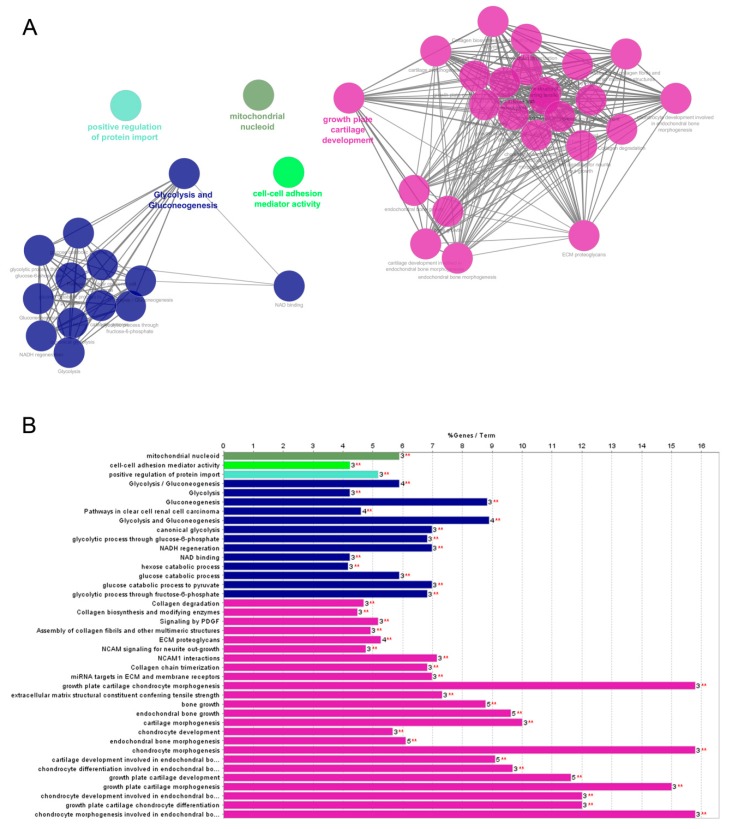
ClueGO + CluePedia analyses of identified deregulated proteins in OC chondrocytes. Analysis was performed integrating biological processes, molecular functions, cellular component, Reactome, Kyoto Encyclopedia of Genes and Genomes (KEGG), and WikiPathways. (**A**) Functionally grouped network with terms as nodes linked based on their kappa score level (≥0.4), where only the label of the most significant term per group is shown; (**B**) GO/pathway terms specific for deregulated proteins. The bars represent the number of proteins associated with the terms. The percentage of genes per term is shown as bar label. Term/group resulted over significance ** *p* < 0.001.

**Figure 4 ijms-20-06179-f004:**
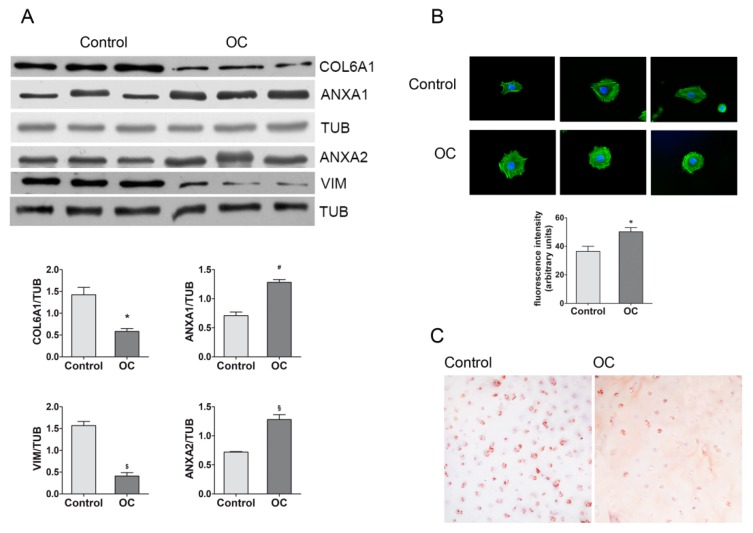
Validation of a panel of five proteins found to be modulated in equine OC cells using different approaches. (**A**) Representative images of Wester Blotting of vimentin (VIM), annexin A2 and A1 (ANXA1-2), and collagen VI alpha 1 chain (COL6A1), as well as tubulin (TUB), which were used as housekeeping proteins. The bar graph shows normalized Western blot band densities. Images of independent blots were acquired and quantified using Quantity One Software. Data represent the mean ± SEM of three independent experiments. (* *p* = 0.01; # *p* = 0.0018; $ *p* = 0.001; § *p* = 0.0027). (**B**) Actin analysis by fluorescent phalloidine of Control and OC cells (scale bar 50 μm), seeded on glass slips. Data represent the mean ± SEM of 3 independent experiments. (**C**) Immunohistochemistry of equine cartilage samples (from the lateral trochlear ridge of femur) for COL6A1 (characterized by pericellular immunolabeling) (magnification 200×).

**Figure 5 ijms-20-06179-f005:**
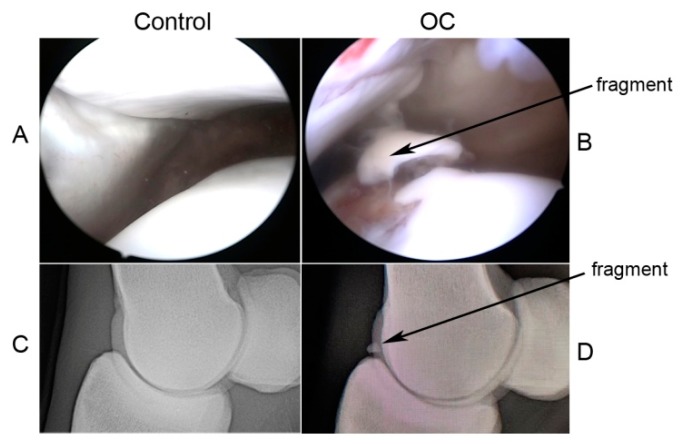
Representative images of healthy (Control) and OC equine joint. (**A**,**B**) Control and OC lateral trochlear ridge of the femur with fragment obtained during arthroscopy. (**C**,**D**) Lateromedial radiographic images of healthy (Control) and OC metacarpophalangeal joint with OC fragment of the dorsoproximal aspect of the proximal phalanx.

**Table 1 ijms-20-06179-t001:** Differentially represented protein spots in OC chondrocytes. Spot number (marked and numbered in Figure 1), protein name, SwissProt accession, theoretical and experimental pI and Mr values, fold changes, and relative *p* values are listed. Asterisk indicates proteins identified as probable fragments. “+++” indicates spots present only in OC gels.

N° Spot	Protein Name	UniProt Accession	Gene Name	pI/Mw Theor(kDa)	pI/Mw Exp(kDa)	Fold Change	*p* Value
1	Collagen alpha-1(VI) chain	F6UW03_HORSE	*COL6A1*	5.24/110	4.55/105	0.44	0.035
2	Collagen alpha-1(VI) chain	F6UW03_HORSE	*COL6A1*	5.24/110	4.68/105	0.40	0.035
3	Collagen alpha-1(VI) chain	F6UW03_HORSE	*COL6A1*	5.24/110	4.96/105	0.42	0.0001
4	Collagen alpha-1(VI) chain	F6UW03_HORSE	*COL6A1*	5.24/110	5.05/105	0.36	0.0001
5	Collagen alpha-1(VI) chain	F6UW03_HORSE	*COL6A1*	5.24/110	5.11/105	0.30	0.0001
6	Collagen alpha-1(VI) chain	F6UW03_HORSE	*COL6A1*	5.24/110	5.18/105	0.49	0.005
7	Collagen alpha-1(VI) chain	F6UW03_HORSE	*COL6A1*	5.24/110	5.25/106	0.41	0.045
8	Endoplasmin	F6ZUJ2_HORSE	*HSP90B1*	4.76/92.6	4.71/83.2	0.44	0.045
9	Collagen alpha-3(VI) chain *	F6QAT0_HORSE	*COL6A3*	6.07/342.2	5.27/95	+++	
10	Collagen alpha-3(VI) chain *	F6QAT0_HORSE	*COL6A3*	6.07/342.2	5.33/95	+++	
11	Collagen alpha-3(VI) chain *	F6QAT0_HORSE	*COL6A3*	6.07/342.2	5.39/95.2	+++	
12	Collagen alpha-3(VI) chain *	F6QAT0_HORSE	*COL6A3*	6.07/342.2	5.45/95.2	+++	
13	Collagen alpha-3(VI) chain *	F6QAT0_HORSE	*COL6A3*	6.07/342.2	5.68/82.4	3.01	0.011
14	Collagen alpha-3(VI) chain *	F6QAT0_HORSE	*COL6A3*	6.07/342.2	5.78/81.8	4.02	0.017
15	Collagen alpha-3(VI) chain *	F6QAT0_HORSE	*COL6A3*	6.07/342.2	5.87/81.8	3.64	0.002
16	Collagen alpha-3(VI) chain *	F6QAT0_HORSE	*COL6A3*	6.07/342.2	6.00/81.7	2.73	0.031
17	Collagen alpha-2(VI) chain	F7CGV8_HORSE	*COL6A2*	5.93/105.6	6.41/81.4	0.48	0.009
18	Collagen alpha-2(VI) chain	F7CGV8_HORSE	*COL6A2*	5.93/105.6	6.65/80.8	0.46	0.0001
19	Collagen alpha-2(VI) chain	F7CGV8_HORSE	*COL6A2*	5.93/105.6	6.79/80.6	0.47	0.002
20	Collagen alpha-2(VI) chain	F7CGV8_HORSE	*COL6A2*	5.93/105.6	7.11/80.4	0.28	0.0001
21	Collagen alpha-2(VI) chain	F7CGV8_HORSE	*COL6A2*	5.93/105.6	7.33/80.1	0.36	0.001
22	Collagen alpha-2(VI) chain	F7CGV8_HORSE	*COL6A2*	5.93/105.6	7.39/77.1	0.15	0.0001
23	Collagen alpha-2(VI) chain	F7CGV8_HORSE	*COL6A2*	5.93/105.6	7.66/76.6	0.25	0.0001
24	Collagen alpha-2(VI) chain	F7CGV8_HORSE	*COL6A2*	5.93/105.6	7.94/75.6	0.20	0.0001
25	Collagen alpha-2(VI) chain	F7CGV8_HORSE	*COL6A2*	5.93/105.6	8.20/75.3	0.42	0.0001
26	Vimentin	F7B5C4_HORSE	*VIM*	4.67/42.2	5.18/74.3	0.28	0.005
27	Vimentin	F7B5C4_HORSE	*VIM*	4.67/42.2	5.29/73.9	0.40	0.003
28	Lactadherin	F7B0S3_HORSE	*MFGE8*	6.22/46.4	6.06/65.2	2.76	0.014
29	Lactadherin	F7B0S3_HORSE	*MFGE8*	6.22/46.4	6.70/64.2	2.10	0.0002
30	ATP synthase subunit beta	F6U187_HORSE	*ATP5F1B*	5.15/56.2	5.24/55.5	3.15	0.017
31	ATP synthase subunit beta	F6U187_HORSE	*ATP5F1B*	5.15/56.2	5.16/54.5	5.90	0.038
32	Hyaluronan and proteoglycan link protein 1	HPLN1_HORSE	*HAPLN1*	8.19/40. 7	5.13/50.1	6.59	0.001
33	Endoplasmic reticulum resident protein 44	F6WY40_HORSE	*ERP44*	5.00/47.44	5.25/50.9	2.27	0.005
34	Trifunctional enzyme mitochondrial.	F6TCZ6_HORSE	*HADHB*	9.49/51.4	5.36/50.9	7.27	0.0001
35	Collagen alpha-1(VI) chain	F6UW03_HORSE	*COL6A1*	5.24/110	5.45/50.8	3.60	0.006
36	Junction plakoglobin *	F6V1T9_HORSE	*JUP*	5.73/82.5	5.54/50.9	6.64	0.004
37	Enolase 1	F6V7C1_HORSE	*ENO1*	6.37/47.5	5.63/51.0	2.76	0.007
38	Ornithine aminotransferase	F7C9Y5_HORSE	*OAT*	6.84/48.7	7.45/50.4	0.29	0.0001
39	Adenosylhomocysteinase	F6VIG9_HORSE	*AHCY*	5.86/48.1	6.90/48.7	0.21	0.009
40	Adenosylhomocysteinase	F6VIG9_HORSE	*AHCY*	5.86/48.1	7.15/48.4	0.15	0.001
41	Cartilage intermediate layer protein	F7C2J3_HORSE	*CILP*	8.85/135	7.37/48.3	0.37	0.0001
42	Actin, cytoplasmic 1	F7AAK7_HORSE	*ACTG1*	5.31/42.1	5.51/43.2	11.9	0.0001
43	Actin, cytoplasmic 1	F7AAK7_HORSE	*ACTG1*	5.31/42.1	5.60/43.3	4.22	0.008
44	Hyaluronan and proteoglycan link protein 1	HPLN1_HORSE	*HAPLN1*	8.19/40.6	5.66/42.7	8.64	0.0005
45	Hyaluronan and proteoglycan link protein 1	HPLN1_HORSE	*HAPLN1*	8.19/40.6	5.26/40.1	0.09	0.004
46	Hyaluronan and proteoglycan link protein 1	HPLN1_HORSE	*HAPLN1*	8.19/40.6	5.45/40.0	0.23	0.0006
47	Collagen alpha-2(VI) chain	F7CGV8_HORSE	*COL6A2*	5.93/105.6	4.94/37.9	0.42	0.012
48	Annexin A2	F6ZI51_HORSE	*ANXA2*	6.92/38.8	7.21/37.9	2.00	0.0023
49	Annexin A1	ANXA1_HORSE	*ANXA1*	6.57/39	7.42/38.0	2.02	0.0002
50	Glyceraldehyde-3-phosphate dehydrogenase	F6YV40_HORSE	*GAPDH*	8.22/36.0	7.75/38.2	3.11	0.0001
51	L-lactate dehydrogenase	C6L1J5_HORSE	*LDHB*	5.85/36.8	6.58/33.9	2.93	0.004
52	Collagen alpha-1(VI) chain	F6UW03_HORSE	*COL6A1*	5.24/110.	7.01/33.7	2.8	0.0004
53	Succinyl-CoA ligase	F6QTC9_HORSE	*SUCLG1*	9.27/36.6	7.28/33.2	2.22	0.0003
54	Voltage dependent anion channel 2	F6TLU0_HORSE	*VDAC2*	7.60/32.3	7.62/33.7	4.01	0.0001
55	14-3-3 protein theta	F6SP02_HORSE	*YWHAQ*	4.68/28.0	4.72/30.4	0.39	0.015
56	Triosephosphate isomerase	F6TZS9_HORSE	*TPI1*	6.96/26.9	8.10/28.2	2.98	0.0001
57	Enoyl-CoA hydratase	F6QV70_HORSE	*ECHS1*	7.55/28.2	8.64/27.1	3.49	0.0001
58	Superoxide dismutase	F6U991_HORSE	*HADHB*	8.44/24.4	7.24/26.4	4.39	0.0001
59	Superoxide dismutase	F6U991_HORSE	*HADHB*	8.44/24.4	7.75/25.5	0.27	0.0001
60	Superoxide dismutase	F6U991_HORSE	*HADHB*	8.44/24.4	8.11/25.4	0.23	0.0001
61	Collagen alpha-1(VI) chain *	F6UW03_HORSE	*COL6A1*	5.24/110	7.82/23.1	+++	
62	Collagen alpha-1(VI) chain *	F6UW03_HORSE	*COL6A1*	5.24/110	7.81/22.4	+++	
63	Collagen alpha-1(VI) chain *	F6UW03_HORSE	*COL6A1*	5.24/110	8.11/22.7	+++	
64	Collagen alpha-1(VI) chain *	F6UW03_HORSE	*COL6A1*	5.24/110	8.11/22.2	+++

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
