# Peer review of "Proteome Alterations in Equine Osteochondrotic Chondrocytes"

_ijms, 2019, doi:10.3390/ijms20246179_

Round 1
Reviewer 1 Report
Proteome alterations in osteochondrotic chondrocytes
By Elisabetta Chiaradia et al.,
OC is indeed still an entity of cartilage disorders with unknown pathogenesis which is so far under-investigated and plays in humans and horses a role. The approach of the manuscript combining bioinformatics and in situ analyses sounds interesting. Some methological information is lacking.
title
insert: „equine“
abstract
line 12 vs line 15: introduce „OC“ as osteochondrosis adequately
„chondrocyte biogenesis impairment“ it is not clear what this term should mean…
„performed on osteochondrotic and healthy chondrocytes“ species? From which joint?
Line 43: „the molecular events than (correct: that) can drive“
Lacking blanks: e.g. line 35, line 151, line 212 and 218
Table 1: where is „figure x“? fold changes: what does „+++“ mean, was there no value?
Please bring the headings on 2 lines.
Legend Fig. 1 „IPG“ explain the abbreviation
Line 121: „correct: „…-corresponding“
2.3.: explain why especially VIM, ANX1 and ANX2 were selected
Line 139 versus line 141: „tubulin“ versus „TUB“ select a consistent writing style
Fig. 2: the labeling of the nodes is too small
Fig. 3: explain what the red asterisk mean. The legend should provide more information.
Fig. 4 B: are cultured cells shown, i fit should depict articular cartilage in situ: which zone? Mention the species in the legend of the figure.
Line 181: surplus point.
Line 189: cytoskeletal alterations: describe which changes were observed.
4.1: provide the gender of the donor animals
Legend fig. 5: mention the donor species.
Line 258: cartilage surface, mention what it means, which zone?
Discussion
Line 192: „VIM, which are both involved in chondrogenesis and chondrocyte differentiation“ vimentin is a mesenchymal intermediate filament also expressed in completely undifferentiated mesenchymal stromal cells and many other mesenchymal cells
Line 202: Further modifications of the cytoskeleton. The results of phalloidin staining (Fig. 4) could be discussed here.
Methods
4.5.3
Line 338: which antibodies were used?
How were the chondrocytes isolated?
Author Response
OC is indeed still an entity of cartilage disorders with unknown pathogenesis which is so far under-investigated and plays in humans and horses a role. The approach of the manuscript combining bioinformatics and in situ analyses sounds interesting. Some methological information is lacking.
Answer: We thank the Reviewer for comments and for the critical revision. We appreciate it and believe that it helped us to improve our manuscript. We have provided a point-by-point reply to all the requests made.
title
insert: „equine“
Answer: The term "equine" has been included in the title according to the Reviewer’s suggestion
abstract
line 12 vs line 15: introduce „OC“ as osteochondrosis adequately
chondrocyte biogenesis impairment“ it is not clear what this term should mean…
Line 43: „the molecular events than (correct: that) can drive“
Lacking blanks: e.g. line 35, line 151, line 212 and 218.....
Answer: Text has been amended following the Reviewer’s suggestions
„performed on osteochondrotic and healthy chondrocytes“ species? From which joint?,
Answer: The species has been added. Unfortunately, we cannot add the term joint to avoid the overcoming of the words limits. This term was extensively cited in the text.
Table 1: where is „figure x“? fold changes: what does „+++“ mean, was there no value?
Answer: The number of the figure has been corrected. "+++" indicates spots present only in OC gels, and for whom it is impossible to calculate fold change values. Table description has changed accordingly.
Please bring the headings on 2 lines.
Legend Fig. 1 „IPG“ explain the abbreviation
Line 121: „correct: „…-corresponding“
Answer: Text has been modified accordingly
2.3.: explain why especially VIM, ANX1 and ANX2 were selected.
Answer: We selected vimentin, ANX1 and ANX2 to validate the quantitative changes impacting alterations of cytoskeleton, also assayed by florescence analysis. Vimentin is the main protein of the intermediate filaments, while ANX1 and ANX2 are both involved in actin cytoskeletal rearrangements. Following the Reviewer’s suggestion, a brief explanation as been included in the text. On the other hand, ANX1 and ANX2 were chosen also for their involvement in cellular functions already described as altered in OC, i.e. apoptosis. This has also been included in the Discussion.
Line 139 versus line 141: „tubulin“ versus „TUB“ select a consistent writing style
Answer: Text has been amended accordingly
Fig. 2: the labeling of the nodes is too small
Fig. 3: explain what the red asterisk mean. The legend should provide more information.
Fig. 4 B: are cultured cells shown, i fit should depict articular cartilage in situ: which zone? Mention the species in the legend of the figure.
Answer: Following the Reviewer’s suggestions, the size of figures has been increased and the legend of Figure 3 and 4 has been amended. More information has been included.
Line 181: surplus point.
Answer: Points have been cancelled.
Line 189: cytoskeletal alterations: describe which changes were observed.
Answer: The deregulation of proteins involved or related to cytoskeleton has been described in the following sentences and paragraphs. eg. ".... In particular, changes were observed in ACTG1 and VIM ....."
4.1:
provide the gender of the donor animals -
Legend fig. 5: mention the donor species.
Line 258: cartilage surface, mention what it means, which zone?
Answer: Text and legend have been modified according to Reviewer's suggestions. Lines 257-276
Discussion
Line 192: VIM, which are both involved in chondrogenesis and chondrocyte differentiation“ vimentin is a mesenchymal intermediate filament also expressed in completely undifferentiated mesenchymal stromal cells and many other mesenchymal cells.
Answer: We agree with the Reviewer that vimentin is the main protein of intermediate filaments that is found in various non-epithelial cells, especially mesenchymal cells. However, in this study we discussed the possible consequences of quantitative changes of vimentin as related to chondrocyte and joint diseases.
Line 202: Further modifications of the cytoskeleton. The results of phalloidin staining (Fig. 4) could be discussed here.
Answer: Following the Reviewer’s suggestions, discussion has been modified as requested.
Methods
4.5.3
Line 338: which antibodies were used?
Answer: For Immunohistochemical evaluation of COL6A1 in cartilage, we used antibodies against COL6A1 (Santa Cruz Biotechnology). This was included in the text.
For actin labeling by Fluorescently-labeled phalloidin, we used Alexa FluorVR 488 Phalloidin that is conjugated to green-fluorescent Alexa Fluor® 488 dye; other antibodies were not necessary.
How were the chondrocytes isolated?.
Answer: This information has been added in this amended version.
Reviewer 2 Report
Chiaradia et al. reported in their manuscript the differentially expressed proteins in chondrocytes isolated from osteochondrotic cartilage fragments. The experiments were well designed, and the results were fully analysed, validated, and properly interpreted. The paper is informative and well written and structured. I do not have further comments except font size of Fig. 2A, 3A and 4A (bar-graphs) needs to be enlarged.
Author Response
Chiaradia et al. reported in their manuscript the differentially expressed proteins in chondrocytes isolated from osteochondrotic cartilage fragments. The experiments were well designed, and the results were fully analysed, validated, and properly interpreted. The paper is informative and well written and structured. I do not have further comments except font size of Fig. 2A, 3A and 4A (bar-graphs) needs to be enlarged.
Answer: We thank the Reviewer for positive comments. We have provided to enlarge figures, as suggested.
Round 2
Reviewer 1 Report
My previous comments have been fully addressed.
For the proof:
Line 155: „florescence analysis“ correct: fluorescence